# Dietary Supplementation with *Dunaliella Tertiolecta* Prevents Whitening of Brown Fat and Controls Diet-Induced Obesity at Thermoneutrality in Mice

**DOI:** 10.3390/nu12061686

**Published:** 2020-06-05

**Authors:** Yukari Yamashita, Tamaki Takeuchi, Yuki Endo, Ayumi Goto, Setsuko Sakaki, Yuji Yamaguchi, Hiroyuki Takenaka, Hitoshi Yamashita

**Affiliations:** 1Department of Biomedical Sciences, College of Life and Health Sciences, Chubu University, Kasugai 487-8501, Japan; yappy8008@na.commufa.jp (Y.Y.); takeuchi@isc.chubu.ac.jp (T.T.); yendo@isc.chubu.ac.jp (Y.E.); a-goto@isc.chubu.ac.jp (A.G.); 2MAC Gifu Research Institute, MicroAlgae Corporation, 4-15 Akebono, Gifu 500-8148, Japan; sakaki@mac-bio.co.jp (S.S.); yamaguchi@mac-bio.co.jp (Y.Y.); takenaka@mac-bio.co.jp (H.T.)

**Keywords:** brown fat, fibroblast growth factor-21, microalga, obesity, thermoneutrality, uncoupling protein 1

## Abstract

We investigated the effect of evodiamine-containing microalga *Dunaliella tertiolecta* (DT) on the prevention of diet-induced obesity in a thermoneutral C57BL/6J male (30 °C). It attenuates the activity of brown adipose tissue (BAT), which accelerates diet-induced obesity. Nine-week-old mice were fed a high-fat diet supplemented with 10 g (Low group) or 25 g (High group) DT powder per kg food for 12 weeks. Compared to control mice without DT supplementation, body weight gain was significantly reduced in the High group with no difference in food intake. Tissue analyses indicated maintenance of multilocular morphology in BAT and reduced fat deposition in liver in DT-supplemented mice. Molecular analysis showed a significant decrease in mammalian target of rapamycin−ribosomal S6 protein kinase signaling pathway in white adipose tissue and upregulation in mRNA expression of brown fat-associated genes including fibroblast growth factor-21 (*Fgf21*) and uncoupling protein 1 (*Ucp1*) in BAT in the High group compared to the control. In the experiments using C3H10T1/2 adipocytes, DT extract upregulated mRNA expression of brown fat-associated genes in dose-dependent and time-dependent manners, accompanied by a significant increase in secreted FGF21 levels. Our data show the ability of DT as a nutraceutical to prevent brown fat attenuation and diet-induced obesity in vivo.

## 1. Introduction

Adipose tissues are largely responsible for lipid and glucose metabolism, affecting energy homeostasis under the control of various hormones and cytokines in the body. Obesity is characterized by excess fat deposition mainly in white adipose tissue (WAT) and subsequently leads to fat accumulation in the liver, which is a serious health risk in industrialized societies [1]. In contrast to WAT, brown adipose tissue (BAT) can undergo thermogenesis mediated by mitochondrial uncoupling protein 1 (UCP1), which maintains homeothermy in mammals in a cold environment [2,3]. As UCP1-mediated thermogenesis dissipates caloric energy as heat, BAT also plays a crucial role in energy metabolism. UCP1 deficiency increases susceptibility to diet-induced obesity (DIO) with age in mice [4]. Adult humans have functional BAT, but it was found to decrease with age and showed an inverse correlation with the degree of adiposity [5]. This underscores the importance of considering BAT as a potential tissue in the treatment of obesity and metabolic syndrome [6,7]. Moreover, discovery of endocrine factors such as fibroblast growth factor 21 (FGF21), which induces UCP1-expressing brown-like adipocytes (beige adipocytes) in WAT, have accelerated basic and clinical studies on the stimulators of brown/beige fat formation and activity [7,8].

Microalgae are good sources of various nutraceuticals such as vitamins, carotenoids, polyunsaturated fatty acids, sterols, and proteins [9]. These constituents from *Spirulina*, *Chlorella*, *Dunaliella*, *Scenedesmus*, and *Phaeodactylum* have been reported to have various health benefits including anti-oxidant, anti-inflammatory, and anticancer properties. They also help reduce blood pressure, cholesterol, and body weight [9,10]. Kumar et al. reported that supplementation of *Scenedesmus dimorphus* and *Schroederiella apiculata* in rat attenuates obesity and its associated pathology such as glucose intolerance and fatty liver [11]. Gille et al. have recently demonstrated that daily oral injection of ethanolic extract of *Phaeodactylum tricornutum* improved diet-induced obesity in mice at 22 °C [12]. This study demonstrated an upregulation of *Ucp1* mRNA in inguinal WAT (IWAT) and an increase in its protein level in interscapular BAT (IBAT) in mice administered with the microalga extract compared to those in control mice, which could be caused by the effect of fucoxanthin.

Evodiamine (Evo) is a major alkaloid compound extracted from the fruit of *Evodia fructus* (*Evodia rutaecarpa* Benth., Rutaceae), which has been used as a traditional Chinese herbal medicine for the treatment of pain, vomiting, and pyresis. Numerous studies have demonstrated that Evo exhibits anti-nociceptive, anti-obesity, anti-tumor, anti-inflammatory, and vasodilatory effects [13,14,15]. We have also demonstrated that it prevents body weight gain in diet-induced, age-related, and genetic obesity in mice [16,17,18]. In these studies, we found that Evo inhibits insulin-stimulated mammalian target of rapamycin (mTOR)-ribosomal S6 protein kinase (S6K) activation in 3T3-L1 adipocytes and WAT, a mechanism which contributes to the improvement of obesity and insulin resistance in obese models. Thus, Evo shows a great potential for the treatment of metabolic diseases including obesity and diabetes.

Recently, we discovered several species of microalgae containing evodiamine, of which *Dunaliella tertiolecta* (DT) produced the highest level of Evo [19]. DT is a strictly photoautotrophic, unicellular chlorophyte with a single cup-shaped chloroplast and a photosynthetic apparatus similar to that of higher plants [20]. DT has been reported to exhibit skeletal muscle relaxant activity [21], plasma cholesterol-reducing activity [22], and antiaggregant activity [23]. However, whether this microalga exhibits anti-obesity effect has not yet been investigated. As DT contains Evo, this green microalga may prevent obesity. Therefore, in the present study, we aimed to clarify the potential of this microalga as a food supplement for prevention of obesity in mice. In addition, we performed animal experiments under thermoneutral condition at 30 °C, in which BAT thermogenesis including UCP1 expression is greatly attenuated, increasing susceptibility to diet-induced obesity, and thereby leading us to strictly assess the effect of DT on brown fat formation and obesity. This is the first report showing the effects of DT on FGF21 production, brown fat formation, and inhibition of diet-induced obesity.

## 2. Materials and Methods 

### 2.1. Microalgae Cultivation and Preparation of Algal Extract

DT and *Dunaliella salina* (DS) were grown in outdoor open raceway ponds at the MAC Miyako farm in Japan as previously described [19]. Sample preparation and evodiamine quantification were performed according to our protocol [19]. In the preparation of DT and DS extracts, 6 g of dried biomasses were extracted with 100 mL ethanol in a reflux device at 90 °C for 1 h. After the extracts were centrifuged at 3200 rpm for 5 min, the supernatants were filtered using filter papers. The filtrates were then concentrated to a final volume of 50 mL by a rotary evaporator apparatus at 90 °C. The Evo content in the extract was estimated to be 0.0474 and 0.007 μg/mL in DT and DS, respectively.

### 2.2. Experimental Animals

Inbred C57BL/6J mice were maintained under artificial lighting for 12 h per day and provided with standard chow (11.6% fat; Diet No. CE-2, CLEA Japan, Inc., Shizuoka, Japan) and tap water ad libitum in our animal facility at 30 °C. Nine-week-old male C57BL/6J mice were divided randomly into three groups and were fed a high-fat diet (HFD) (41.9% fat; Diet No. D15040, CLEA Japan, Inc.) with 0 g, 10 g, or 25 g DT powder per kg food (corresponding to 0, 4 and 10 μg Evo per kg food, respectively) for 12 weeks (Figure 1). These mice were housed individually. Body weight and food intake were measured once per week. Tissue and blood samples were collected and stored at −80 °C or were fixed in 4% paraformaldehyde dissolved in phosphate-buffered saline (PBS). All animal experiments were performed in strict accordance with the recommendations of the Fundamental Guidelines for Proper Conduct of Animal Experiments and Related Activities in Academic Research Institutions under the jurisdiction of the Ministry of Education, Culture, Sports, Science, and Technology, Japan. The protocol was approved by the Institutional Animal Care and Use Committee of Chubu University (#2810013).

### 2.3. Blood Biochemistry

Blood samples were collected from the tail vein and immediately used to determine glucose level using a glucometer (NovoAssist Plus, Novo Nordisk, Tokyo, Japan). Serum insulin level was measured using enzyme-linked immunosorbent assay kits (Lebis-insulin-mouse, Sibayagi, Gunma, Japan). Serum triglyceride (TG) and total cholesterol (TC) levels were measured using TG E-test and T-Cho E-test kits (Wako Pure Chemicals, Osaka, Japan), respectively. Total liver lipids were extracted using the previously described protocol with minor modifications [24]. Briefly, liver samples (100 mg) were homogenized in 1 mL of hexane/isopropyl alcohol (3:2, *v*/*v*). The homogenized samples were centrifuged, and the supernatant was evaporated under reduced pressure. The dried samples were then resuspended in 10% triton X-100 prepared in isopropyl alcohol, and the lipid levels were assayed using commercial kits. FGF21 levels in serum and culture medium were measured using mouse FGF-21 DuoSet ELISA (R&D SYSTEMS, Minneapolis, MN, USA).

### 2.4. C3H10T1/2 Adipocyte Culture

C3H10T1/2-clone 8 cells (#IF050415) were obtained from Health Science Research Resources Bank (Osaka, Japan). Cells were inoculated into multi-well plates and grown in Dulbecco’s Modified Eagle Medium (DMEM, Wako Pure Chemicals) containing 10% calf serum (Biowest, Nuaillé, France) at 37 °C in 5% CO_2_. Once confluent, the cells were differentiated by maintaining them in DMEM supplemented with 10% fetal bovine serum (FBS, ICN, Irvine, CA, USA), 10 μg/mL insulin, 1 μM dexamethasone, 0.5 mM 3-isobutyl-1-methylxanthine, 125 μM indomethacin, and 3 nM T3 for 2 days, and then in 10% FBS/DMEM containing insulin and T3 for 5 days. On day-7 after differentiation, the cells were washed with serum-free DMEM and treated with DT extract or DS extract in the absence of above stimulators and serum for different time periods indicated in time course experiments and 18 h for all the other experiments.

### 2.5. Protein Analysis

Western blot analyses were performed on C3H10T1/2 cell lysates and total tissue lysates of epididymal WAT (EWAT), IBAT, and liver as described previously [16]. The concentrations of protein in the lysates were measured using BCA protein assay (Pierce Biotechnology, Rockford, IL, USA). Proteins were separated on 4–20% SDS-polyacrylamide gels (Daiichi Pure Chemicals, Tokyo, Japan) and transferred onto Immobilon polyvinylidene difluoride membranes (Millipore, Bedford, MA, USA). The membranes were incubated with specific antibodies against mTOR, phospho-Ser2448 mTOR, p70S6 kinase, phospho-Thr389 p70S6 kinase, ribosomal protein S6 (rpS6), phospho-Ser235/236 rpS6, α/β-tubulin (all from Cell Signaling Technology, Danvers, MA, USA), or UCP1 (ab10983, Abcam, Cambridge, UK). After incubation with the appropriate secondary antibody for 1 h at room temperature, specific signals were detected using the Immobilon western chemiluminescent HRP substrate (Millipore, Billerica, MA, USA). The resulting images were quantified using NIH Image (version 1.63).

### 2.6. Gene Expression Analysis

Total RNA was extracted using TRI REAGENT (Molecular Research Center, Inc., Cincinnati, OH, USA), according to the manufacturer’s protocol. To quantify mRNA expression levels, RNAs from tissues or cultured cells were reverse-transcribed using High Capacity cDNA Reverse Transcription kits (Applied Biosystems, Foster City, CA, USA), as per manufacturer’s instructions, and real-time RT-PCR analysis was performed using a Light-Cycler and THUNDERBIRD SYBR qPCR Mix (TOYOBO, Osaka, Japan). All gene expression data were normalized relative to 36B4 levels. The oligonucleotide primer sequences are mentioned in Table 1.

### 2.7. Histological Analysis

Tissue samples were removed and immediately fixed by immersion in 10% formaldehyde prepared in neutral buffer solution (Kishida Chemical, Osaka, Japan) at 4 °C. They were then dehydrated, cleared, and paraffin-embedded. Paraffin sections (4 μm) were stained with hematoxylin and eosin.

### 2.8. Statistical Analysis

Data are expressed as mean ± standard error of the mean (SE). Statistical analyses were performed using the StatView (SAS Institute) programs. Significance in differences among more than two groups (from three to six in our study) were assessed using one-way analysis of variance (ANOVA). Two-Way Repeated Measures ANOVA followed by protected Fisher’s Least Significant Difference post-hoc test were applied to determine statistical differences in body weight gain and food intake. Values of *p* < 0.05 were considered statistically significant. 

## 3. Results

### 3.1. Effect of Evodiamine-Containing DT Supplementation on Diet-Induced Obesity at Thermoneutrality in Mice

In comparison to the control group, body weight gain was significantly lower in the High group, but not in the Low group, under HFD and thermoneutral temperature conditions (Figure 2A), although there was no difference in the amount of food consumed among the three groups (Figure 2B). IBAT weight was significantly low and a reduction in EWAT weight was observed in the Low and High groups (*p* = 0.0811 and 0.0558, respectively) compared to those in the control groups (Figure 2C). As expected, histological analysis showed a morphological change in IBAT in the control group at thermoneutrality, where the typical multilocular brown adipocyte morphology changed to white adipocyte-like unilocular morphology, a phenomenon called BAT whitening (Figure 2D). Intriguingly, the multilocular brown adipocyte morphology was maintained in IBAT in case of DT-supplemented mice, especially in the High group, compared to that of the control mice at 30 °C (Figure 2D). In addition, a decrease in adipocyte size was observed in IWAT and EWAT of DT-treated groups compared to that of the control group (Figure 2D). There was no significant difference in serum glucose levels among the three groups (Figure 2E). However, insulin levels tended to decrease in DT-treated groups relative to control (Figure 2F). There were no significant differences in triglyceride (TG) and total cholesterol (TC) among the three groups (Figure 2G), whereas hepatic TG and TC levels were significantly lower in Low and/or High group than the control group (Figure 2H,I), indicating reduced fat deposition in liver in DT-treated mice. These results suggested that DT supplementation improved diet-induced obesity and associated fatty liver in vivo.

### 3.2. Effect of DT Supplementation on Signal Transduction in WAT and Liver

We also examined the effect of DT supplementation on mTOR signaling pathway in EWAT and liver in mice, as DT contains Evo, which is known to inhibit mTOR-S6K signaling in WAT in obese/diabetic mice [17]. As expected, the phosphorylation of mTOR Ser2448 and S6K Thr389 in EWAT significantly decreased in the High group compared to those in the control group and the Low group (Figure 3A,B). In addition, the phosphorylation of rpS6 Ser235/236, which is phosphorylated by S6K, was significantly reduced in EWAT in the High group compared to that in the control group (Figure 3C). On the other hand, there were no differences in the phosphorylation levels of mTOR and S6K in liver among the three groups (Figure 3D,E), whereas a significantly lower phosphorylation level of rpS6 was detected in the High group than the control group (Figure 3F). These results indicated that dietary supplementation of DT with a percentage content of 2.5% downregulated mTOR-S6K signaling in EWAT of mice with diet-induced obesity at thermoneutrality.

### 3.3. DT Supplementation Stimulates Expression of Brown Fat-Associated Genes in IBAT of HFD-Fed Mice at Thermoneutrality

We then examined mRNA expression of brown fat-associated genes in adipose tissues, as the tissue staining indicated that brown adipocyte morphology was well maintained in IBAT of DT-supplemented mice (Figure 2D). In addition to an upregulation of *Ucp1* mRNA (*p* = 0.0628), the mRNA levels of cell death-inducing DFFA-like effector A (*Cidea*), peroxisome proliferator activated receptor-γ coactivator 1-α (*Ppargc1a*), and PR domain containing 16 (*Prdm16*), a dominant regulator of brown fat cell fate [25], were significantly higher in IBAT in the High group than those in the control group (Figure 4A). Moreover, an increase in the *Fgf21* mRNA level was found in IBAT in DT-supplemented mice compared to those of the control mice (Figure 4A). Western blot analysis showed significantly higher UCP1 level in IBAT in the High group than the control group (Figure 4D). On the other hand, no significant differences were observed in these mRNA levels in IWAT and EWAT among the three groups (Figure 4B,C). There were no significant differences in the hepatic *Fgf21* mRNA and serum FGF21 level among the three groups (Figure 4E,F).

### 3.4. DT Extract Stimulates Brown Fat-Associated Gene Expression and FGF 21 Production in C3H10T1/2 Adipocytes

To confirm the stimulatory effect of DT components on brown fat formation, we investigated its effect on the expression of brown fat-associated genes in C3H10T1/2 cells, which is an established in vitro model for brown adipocytes [26]. When adipocytes were treated with DT extract at different concentrations, there was no obvious difference in cell morphology among groups (Figure 5A). However, significant upregulation of *Ucp1*, *Cidea*, and *Prdm16* mRNA expression was found in cells treated with 0.3% DT extract (Figure 5B,C,E). Gene expressions of *Ppargc1a* and *Fgf21* were also upregulated when treated with DT extract, which peaked at the concentration of 1.0% (Figure 5D,F). 

In time course experiments carried out using 0.3% DT extract, the mRNA expressions of *Ucp1* and *Fgf21* increased in a time-dependent manner and the mRNA levels at 24 h were 11.1 and 5.9-fold higher than those at 0 h, respectively (Figure 6A,B). In parallel to *Fgf21* mRNA expression, a remarkable increase of FGF21 level was detected in the conditioned medium (CM) of adipocytes at 24 h post treatment with 0.3% DT extract. However, its level was also elevated without DT stimulation at 24 h (Figure 6C). We then compared the effect of DT on FGF21 production with that of another green alga, DS. Similar to DT, DS extract increased FGF21 production in a dose-dependent manner in C3H10T1/2 adipocytes, where the FGF21 level in CM was significantly higher (1.8-fold) at the concentration of 1.0% DS than the control (Figure 6D). However, DT induced higher FGF21 production (3.2-fold at 1.0%) than that of DS, although its production was significantly inhibited at the concentration of 3.0% extract in both microalgae (Figure 6D).

## 4. Discussion

In this study, we showed for the first time, the effectiveness of a green microalga DT as a nutraceutical for the prevention of diet-induced obesity and hepatic lipid accumulation in vivo without any restriction on food intake. The effects of DT are unique and very potent for the following reasons: Firstly, DT supplementation inhibited mTOR-S6K signaling in visceral WAT of mice, which was similar to the results in our previous studies using Evo diet [17,18] and in the study using rapamycin diet [27]. Suppression of mTOR signaling is thought to mediate major beneficial effects of caloric restriction including suppression of obesity, type 2 diabetes, cancer, and neurodegeneration [28,29]. S6K phosphorylated by mTOR promotes mRNA translation by phosphorylating multiple proteins such as eukaryotic translation initiation factor 4B and rpS6, which is involved in the control of mammalian cell size [30]. Therefore, the significant reduction in rpS6 phosphorylation in WAT and liver by DT supplementation might contribute to suppression of cell growth, consequently contributing to the prevention of lipid accumulation in these tissues of mice.

Secondly, DT supplementation notably inhibited BAT whitening (i.e., attenuation of brown fat) in mice at thermoneutrality. This finding was most intriguing because UCP1 expression and BAT thermogenic function are usually attenuated to a great extent, accelerating susceptibility to obesity in the thermoneutral condition [31,32]. The stimulatory effect of DT for brown fat formation was confirmed in a brown adipocyte culture model, in which the microalga extract stimulated brown fat-selective genes such as *Ucp1* and *Fgf21* in dose-dependent and time-dependent manners. Although we failed to detect an increase of FGF21 level in serum of DT supplemented mice, DT extract significantly increased the secretion of FGF21 in C3H10 adipocyte culture. Interestingly, the effect of DT on FGF21 production was greater than that of DS, which belongs to the same strain. This secreted FGF21 may stimulate brown fat formation, including UCP1 expression in an autocrine/paracrine fashion as reported previously [8]. Véniant et al. reported that FGF21 administration potently prevents whitening and induces UCP1 expression in BAT even at thermoneutrality [33]. FGF21 administration also improves glucose clearance, which requires UCP1 thermogenesis in mice [34]. Although the role of FGF21 appears to be complex because FGF21 performs diverse metabolic functions in multiple target organs [35], the metabolic benefits provided by FGF21 may have pharmacological significance in improving obesity and insulin resistance [36]. Thus, our results suggest that DT administration may stimulate FGF21 induction in BAT and thereby bolster the FGF21 autocrine/paracrine mechanism-mediated differentiation of brown preadipocytes.

Regarding the compounds capable of promoting brown fat formation, carotenoid can be a candidate responsible for the effects of DT. Serra et al. reported that 10 μM concentrations of carotenoids, including β-carotene and lutein, upregulated UCP1 expression in brown adipocytes [37]. β-carotene is enzymatically converted to vitamin A (retinol), and this lipophilic vitamin and its derivatives play an important role in adipocyte biology, due to which this carotenoid has attracted attention in the field of obesity research [38]. Essentially, retinoic acid upregulates UCP1 expression in adipocytes differentiated from mouse embryonic fibroblasts [39], despite blocking adipogenesis in 3T3-L1 preadipocyte, a white adipocyte culture model [40]. However, the concentrations of β-carotene and lutein in DT extract were 13 and 211 μM, respectively, which reduces to 0.039 and 0.633 μM in 0.3% extract. In addition, their concentrations were significantly higher in DS extracts (β-carotene: 3.35 mM, lutein: 580 μM). Therefore, it is plausible that the combination of compounds including carotenoids in DT promotes brown fat formation.

Our findings strongly suggest that DT holds nutraceutical potential for the prevention of obesity and metabolic diseases by regulating adipose tissue functions including stimulation of brown fat formation via FGF21 production. Considering the crucial role of brown fat in energy metabolism and the causal relation between the decline in brown fat activity and obesity progression, DT may serve as a functional food for human health. Further studies may reveal unknown effective compounds involved in brown fat formation and FGF21 production in future study.

## Figures and Tables

**Figure 1 nutrients-12-01686-f001:**
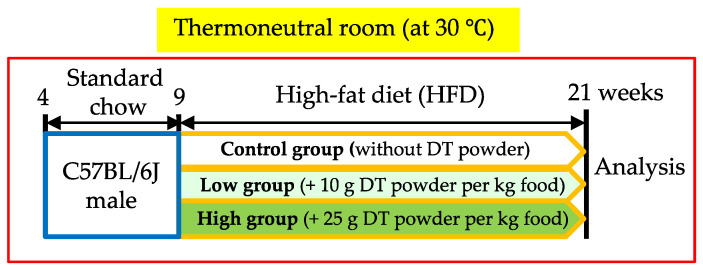
Protocol of animal experiment. Nine-week-old mice were fed the high-fat diet (HFD) supplemented with or without *Dunaliella tertiolecta* (DT) powder for 12 weeks.

**Figure 2 nutrients-12-01686-f002:**
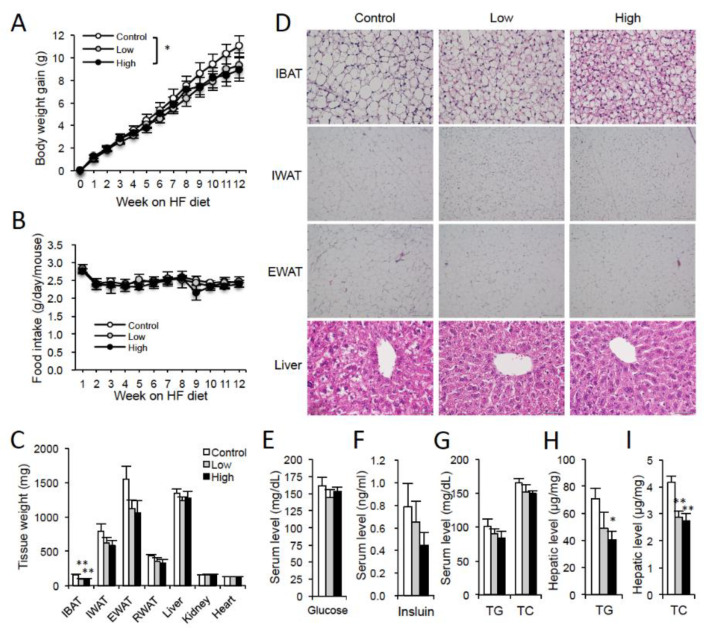
Effect of DT supplementation on diet-induced obesity at thermoneutrality in mice. Nine-week-old male C57BL/6J mice were fed the HFD supplemented with 0 g (Control), 10 g (Low) or 25 g (High) DT powder per kg food for 12 weeks. (**A**) Body weight gain. (**B**) Food intake. (**C**) Tissue weights. IBAT: interscapular BAT, IWAT: inguinal WAT, EWAT: epidydimal WAT, RWAT: retroperitoneal WAT. (**D**) Histology of IBAT, IWAT, EWAT, and liver were analyzed. Tissue sections were stained with hematoxylin and eosin. Representative images are shown. Scale bars, 50 μm for IBAT and liver, 200 μm for IWAT and EWAT. (**E**–**G**) Serum levels of fed glucose (**E**), insulin (**F**), triglyceride: TG, and total cholesterol: TC (**G**) were examined. (**H**,**I**) Hepatic levels of TG (**H**) and TC (**I**) were also analyzed. Data are expressed as the mean ± SE (Control: *n* = 5, Low: *n* = 6, High: *n* = 5). * *p* < 0.05, ** *p* < 0.01 vs. Control.

**Figure 3 nutrients-12-01686-f003:**
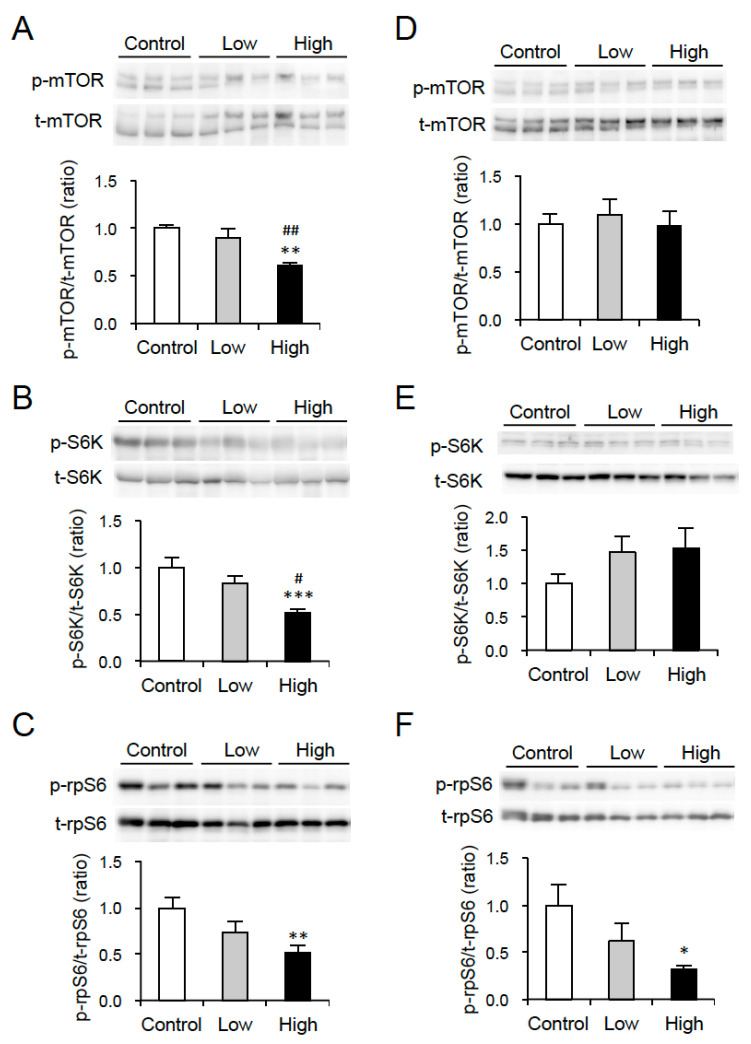
Effect of DT supplementation on mTOR-S6K signaling in WAT and liver of mice. Western blot analyses of mTOR (**A**,**D**), S6K (**B**,**E**), and rpS6 (**C**,**F**) were performed using tissue lysates of EWAT (**A**–**C**) and liver (**D**–**F**) from mice in Figure 2. Representative images are shown. Phosphorylation levels of mTOR Ser2448, S6K Thr389, and rpS6 Ser235/236 were normalized to total level of each protein. Data are expressed as mean ± SE (*n* = 5–6). * *p* < 0.05, ** *p* < 0.01, *** *p* < 0.001 vs. Control. ^#^
*p* < 0.05, ^##^
*p* < 0.01 vs. Low.

**Figure 4 nutrients-12-01686-f004:**
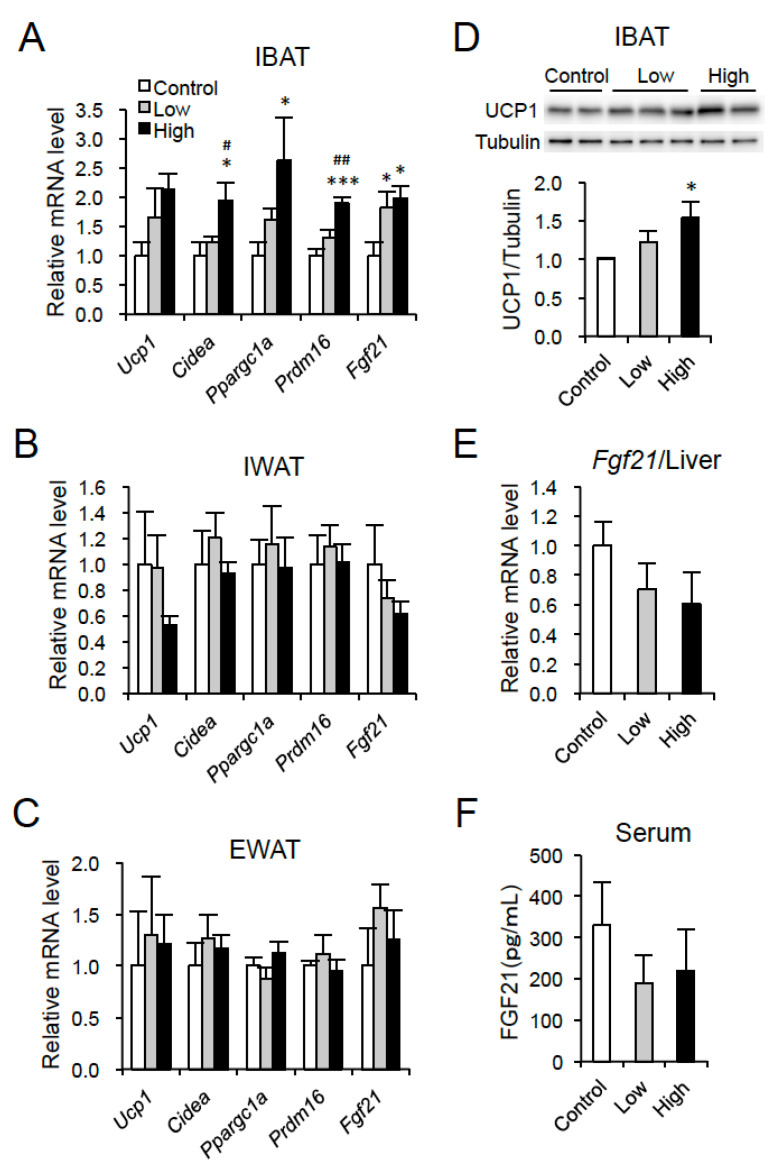
DT supplementation stimulates expression of brown fat-associated genes in IBAT in HFD-fed mice at thermoneutrality. (**A**–**C**) mRNA levels of *Ucp1*, *Cidea*, *Ppargc1a*, *Prdm16*, and *Fgf21* in IBAT (**A**), IWAT (**B**), and EWAT (**C**). (**D**) UCP1 and tubulin levels in IBAT. Representative images are shown. (**E**) *Fgf21* mRNA level in liver. (**F**) Serum FGF21 level. Data are the means ± SE (*n* = 4–6). * *p* < 0.05, *** *p* < 0.001 vs. Control. ^#^
*p* < 0.05, ^##^
*p* < 0.01 vs. Low.

**Figure 5 nutrients-12-01686-f005:**
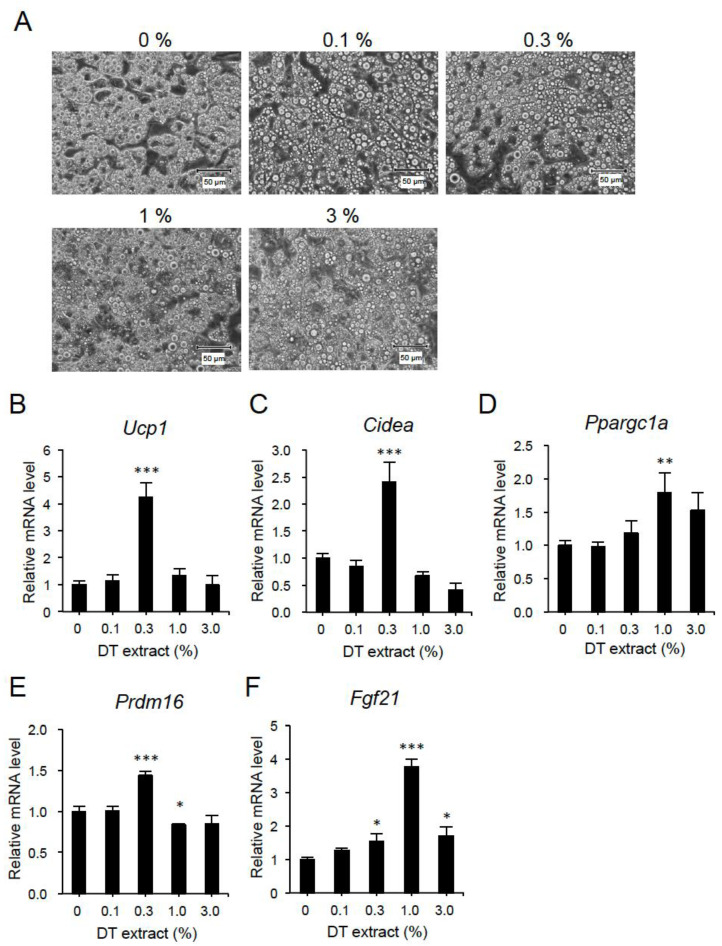
DT extract stimulates expression of brown fat-associated genes in C3H10T1/2 adipocytes. Cells were treated with the indicated concentration of DT extract for 18 h. (**A**) Cell morphology. (**B**–**F**) Dose dependent upregulation of *Ucp1* (**B**), *Cidea* (**C**), *Ppargc1a* (**D**), *Prdm16* (**E**), and *Fgf21* (**F**) mRNAs. (*n* = 7, except *n* = 4 for 3%). Data are the means ± SE. * *p* < 0.05, ** *p* < 0.01, *** *p* < 0.001 vs. 0%.

**Figure 6 nutrients-12-01686-f006:**
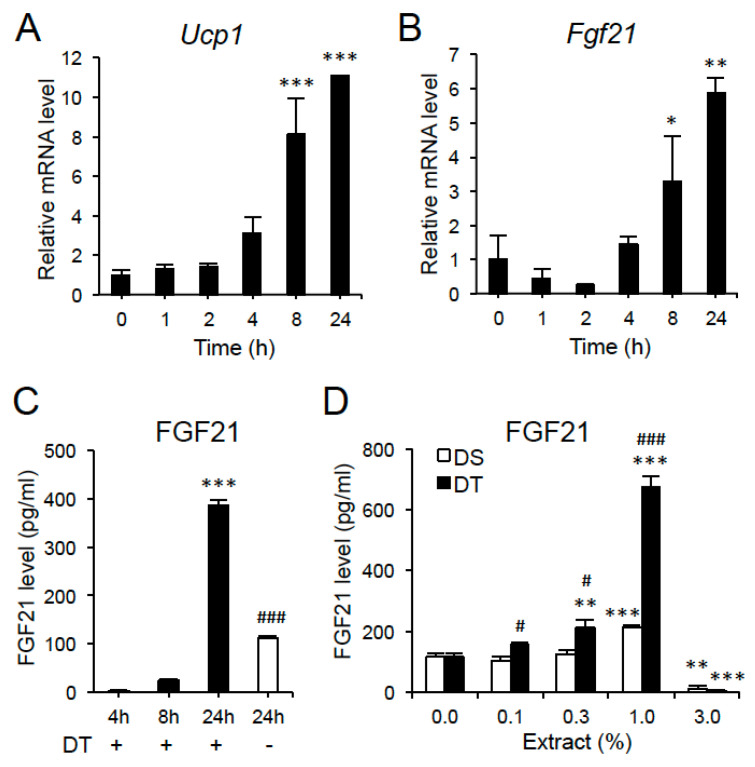
Effects of DT on *Ucp1*, *Fgf21* mRNA expression, and FGF21 production in C3H10T1/2 adipocytes. Cells were stimulated by 0.3% DT extract for the indicated time (**A**–**C**). (**A**) *Ucp1* mRNA level. (**B**) *Fgf21* mRNA level (*n* = 2 for each point, representative of two independent experiments). (**C**) FGF21 level in the conditioned medium (CM) when treated with DT (*n* = 3–5). (**D**) FGF21 level in the CM when treated with DT and DS. Cells were stimulated by the indicated concentration of DT or DS extract for 18 h (*n* = 4). Data are the means ± SE. * *p* < 0.05, ** *p* < 0.01, *** *p* < 0.001 vs. 0 h, 4 h or 0% in the same extract. ^#^
*p* < 0.05, ^###^
*p* < 0.001 vs. DT+ (24 h) or DS extract at same concentration.

**Table 1 nutrients-12-01686-t001:** Real-time RT-PCR primer sequences for mouse genes.

Gene Symbol	Sense	Antisense
*36b4*	TCATCCAGCAGGTGTTTGACA	CCCATTGATGATGGAGTGTGG
*Cidea*	ATCACAACTGGCCTGGTTACG	TACTACCCGGTGTCCATTTCT
*Fgf21*	GTGTCAAAGCCTCTAGGTTTCTT	GGTACACATTGTAACCGTCCTC
*Ppargc1a*	TAGGCCCAGGTACGACAGC	GCTCTTTGCGGTATTCATCC
*Prdm16*	GACATTCCAATCCCACCAGA	CACCTCTGTATCCGTCAGCA
*Ucp1*	GTGAAGGTCAGAATGCAAGC	AGGGCCCCCTTCATGAGGTC

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
