# Peer review of "Dietary Supplementation with Dunaliella Tertiolecta Prevents Whitening of Brown Fat and Controls Diet-Induced Obesity at Thermoneutrality in Mice"

_nutrients, 2020, doi:10.3390/nu12061686_

Round 1
Reviewer 1 Report
I appreciate the opportunity to review this interesting paper on the effect of evodiamine-containing microalga Dunaliella tertiolecta (DT) on the prevention of diet-induced obesity.
The genesis of this paper is the hypothesis that diet enriched with DT could reverse, at least in part, brown adipocyte tissue (BAT) dysfunction in obesity, the so-called whitening of white fat turns the brown adipocyte into a white-like adipocyte. Strategies to increase adaptive thermogenesis have become a promising target for obesity control, so the assumptions of this work seem promising.
The authors have tested this hypothesis on two models.
First, C57BL/6J mice were fed a high-fat diet (HFD) alone as controls or HFD enriched with 10 g or 25 g DT powder per kg food for 12 weeks.
Second C3H10T1/2-clone cells were treated with DT extract for different periods.
They observed that high DT supplementation prevents brown fat dysfunction and diet-induced obesity in vivo. In the experiments using C3H10T1/2 adipocytes, DT extract upregulated mRNA expression of brown fat-associated genes in dose-dependent, which was accompanied by a significant increase in secreted FGF21 levels.
The main strengths of this paper are that it addresses an interesting and timely question, the article is well constructed, the experiments were well conducted, and analysis was well performed.
Considering these strengths, though, as I read the manuscript, I found some areas in which I would have appreciated greater clarity. I believe the paper could be further strengthened by added information:
- A flow chart showing the experimental procedure should be included as it makes the experimental setting more visible to the reader.
- Some details are not entirely clear to me.
- What was the reason for using of Dunaliella salina (DS) extracts? Does it also contain Evodiamine, or does it contain a similar composition to DT except for Evo, and it was used as a control? It is not clear.
- Did the authors consider testing the effect of the test substances on animals on a standard diet?
- Has the distribution of body fat been examined in all groups?
- Did the authors consider possible action of Evo via the vanilloid receptor TRPV1?
Author Response
Reply (point-by-point response) to Reviewer #1
We thank the Reviewer for these positive comments on our study. According to your comments, we revised the original manuscript as below.
----------------------------
Reviewers' comments:
Comments and Suggestions for Authors
I appreciate the opportunity to review this interesting paper on the effect of evodiamine-containing microalga Dunaliella tertiolecta (DT) on the prevention of diet-induced obesity.
The genesis of this paper is the hypothesis that diet enriched with DT could reverse, at least in part, brown adipocyte tissue (BAT) dysfunction in obesity, the so-called whitening of white fat turns the brown adipocyte into a white-like adipocyte. Strategies to increase adaptive thermogenesis have become a promising target for obesity control, so the assumptions of this work seem promising.
The authors have tested this hypothesis on two models.
First, C57BL/6J mice were fed a high-fat diet (HFD) alone as controls or HFD enriched with 10 g or 25 g DT powder per kg food for 12 weeks.
Second C3H10T1/2-clone cells were treated with DT extract for different periods.
They observed that high DT supplementation prevents brown fat dysfunction and diet-induced obesity in vivo. In the experiments using C3H10T1/2 adipocytes, DT extract upregulated mRNA expression of brown fat-associated genes in dose-dependent, which was accompanied by a significant increase in secreted FGF21 levels.
The main strengths of this paper are that it addresses an interesting and timely question, the article is well constructed, the experiments were well conducted, and analysis was well performed.
Considering these strengths, though, as I read the manuscript, I found some areas in which I would have appreciated greater clarity. I believe the paper could be further strengthened by added information:
A flow chart showing the experimental procedure should be included as it makes the experimental setting more visible to the reader.
Some details are not entirely clear to me.
(Response)
According to your suggestion, we added the procedure for animal experiment as Figure 1 in the revised manuscript. In addition, the description of mouse age, two-month-old, was corrected to 9-week-old in the revision.
What was the reason for using of Dunaliella salina (DS) extracts? Does it also contain Evodiamine, or does it contain a similar composition to DT except for Evo, and it was used as a control? It is not clear.
(Response)
As we described in the original manuscript, DS contains evodiamine at a lower level compared to DT (line 92). On the other hand, the concentrations of β-carotene and lutein were much higher in DS extract (3.35 mM and 580 μM, respectively) than in DT extract (13 μM and 211 μM, respectively) (lines 323-326). In the animal experiment, we firstly evaluated the effect of DT, because this microalga produced the highest level of Evo in the microalgae examined (lines 71-72). In the culture experiments using C3H10T1/2 adipocytes, we finally compared the DT effect with the DS effect on FGF21 production, because Serra et al. reported that 10 μM concentrations of carotenoids, including β-carotene and lutein, upregulated UCP1 expression in brown adipocytes (lines 317-319), which could be stimulated by FGF21.
Did the authors consider testing the effect of the test substances on animals on a standard diet?
(Response)
We did not perform the similar experiment on a standard diet, because normal mice hardly get fat by such a low-fat diet even in the obese-prone C57BL/6 strain.
Has the distribution of body fat been examined in all groups?
(Response)
We did not examine the distribution of body fat by using CT image and/or dual-energy X-ray absorptiometry (DEXA), because we do not have these apparatuses. However, our results of adipose tissue weights (Figure 1 in the original manuscript) may suggest a difference of body fat distribution between the control and DT groups; i.e., a reduction in EWAT weight (abdominal fat) in the Low and High groups (lines 179-181 in the original manuscript).
Did the authors consider possible action of Evo via the vanilloid receptor TRPV1?
(Response)
Thank you for this insightful comment. As you pointed out, Evo is an agonist for TRPV1. So, we do not deny a possible action of Evo for brown fat formation. However, we could not detect a positive effect on UCP1 expression in our previous study using long-term dietary supplementation with Evo (Ref. #18). To address your question, it would be required to perform several experiments using TRPV1-KO mice in future study.

Reviewer 2 Report
The article shows the nutraceutical effect of Dunaliella tertiolecta (DT) on obesity, reducing the accumulation of adipose tissue and stimulating brown fat, without changing food intake. DT could be used as a functional food in humans to prevent obesity and the consequences of this pathology in the development of other diseases.
All the sections of the article expressed by the authors are well argued and clearly defined. With which, I only propose small reviews:
- The different subsections of the methodology section do not have a space between them. Lines 120, 131, 144, 153, 158.
- Lines 97-98: Indicate which group corresponds to 0 g, 10 g or 25 g DT powder per kg food; (control, low and high groups).
- Line 305. No questions in the discussion.
Author Response
Reply (point-by-point response) to Reviewer #2
We thank the Reviewer for positive comments on our study. According to your comments, we revised the original manuscript as below.
-----------------------------------------------------
Reviewers' comments:
Comments and Suggestions for Authors
The article shows the nutraceutical effect of Dunaliella tertiolecta (DT) on obesity, reducing the accumulation of adipose tissue and stimulating brown fat, without changing food intake. DT could be used as a functional food in humans to prevent obesity and the consequences of this pathology in the development of other diseases.
All the sections of the article expressed by the authors are well argued and clearly defined. With which, I only propose small reviews:
The different subsections of the methodology section do not have a space between them. Lines 120, 131, 144, 153, 158.
(Response)
As you mentioned, the converted pdf file (nutrients-818713-peer-review.pdf) do not have a space. This may be a converting problem, because our original word file (nutrients-818713) has space properly.
Lines 97-98: Indicate which group corresponds to 0 g, 10 g or 25 g DT powder per kg food; (control, low and high groups).
(Response)
According to your suggestion, we added the procedure for animal experiment as Figure 1 in the revised manuscript. In addition, the description of mouse age, two-month-old, was corrected to 9-week-old in the revision.
Line 305. No questions in the discussion.
(Response)
According to your suggestion, we replaced “What kind of compounds in DT can promote brown fat formation?” with “Regarding the compounds capable of promoting brown fat formation,” in the revised manuscript (line 362).
